# Experimental Investigation on the Effects of Mineral Water Composition on the Leaching of Cement-Based Materials

**DOI:** 10.3390/ma17071548

**Published:** 2024-03-28

**Authors:** Alienor Pouyanne, Sonia Boudache, Benoît Hilloulin, Ahmed Loukili, Emmanuel Roziere

**Affiliations:** 1Nantes Université, Ecole Centrale Nantes, Centre National de la Recherche Scientifique (CNRS), Civil Engineering and Mechanics Research Institute (GeM), Unité Mixte de Recherche (UMR) 6183, 44000 Nantes, France; 2Edycem, Parc d’Activité Vendée Sud Loire, Rue du Fléchet, 85600 Montaigu, France

**Keywords:** leaching, mineral water, mortar, calcium ion, microstructure, 3D microscopy

## Abstract

The common phenomenon observed for concrete in aggressive water is leaching, which involves the dissolution of cement hydration products. Many studies have focused on leaching in demineralised water or acid attacks, but mineral water still deserves further investigation. In most standards, the aggressiveness of a given water body is determined by its pH and not its composition. The effect of the calcium content of the water on degradation is yet to be determined. In this paper, the leaching of Portland cement-based mortar was induced by two types of drinking water with different calcium contents and buffer capacity in controlled conditions. The Langelier saturation index (LSI) was used to describe water aggressiveness based on the calco-carbonic equilibrium. The studied waters had the same pH but LSIs of +0.5 and −1.0 corresponding to scaling with respect to aggressive water; demineralised water was used as a reference. Microstructural damage was checked by TGA and X-ray microtomography. Macroscopic measurements were used to monitor global degradation. The soft water caused a 53% deeper deterioration of the mortar sample than the hard water. Soft water-induced leaching was found to be similar yet slower to leaching via demineralised water (with a mass loss of −2.01% and −2.16% after 200 days, respectively). In contrast, hard water induced strongly time-dependent leaching, and the damage was located close to the surface. The roughness of leached specimens was 18% higher in hard water than in soft water. The formation of calcite on the sample surface not only affects the leaching rate by creating a protective surface layer, but it could also act as a calcium ion pump.

## 1. Introduction

Concrete structures in contact with water deteriorate due to leaching, which affects their serviceability, especially in drinking water treatment plants or dams [1]. Repair works usually require the temporary closure of the water treatment plant, and another plant must, therefore, take over during this period, which might result in considerable deleterious impacts. On the other hand, rehabilitation works usually involve applying a layer of epoxy resin to the surface of the concrete structure, which is extremely expensive and has a negative ecological impact. For this reason, moving towards better water structures and durability relative to leaching is worthwhile.

The leaching phenomenon involves the dissolution of calcium from cement hydration products, such as portlandite, monosulfoaluminates, or C-S-H. In the long term, leaching affects the mechanical strength and the physical properties of cementitious material, such as porosity [2]. For example, Carde and François observed an 11% decrease in compressive strength after a 2-month exposure of concrete to an ammonium nitrate solution [3]. Many numerical and experimental studies have focused on leaching by demineralised water or acid attack [4,5,6,7,8]. However, degradation by mineral water still requires efforts for a better understanding of the physicochemical phenomena at stake [9,10], as ion leaching kinetics may be particularly complex [11]. In recent years, concrete tank deterioration has been observed in several drinking water treatment plants. Since the different processes for water treatment significantly affect the water composition and pH, water in the different tanks of treatment plants cannot be equated to demineralised water. Thus, performance-based specifications and relevant indicators are greatly needed for developing innovative and sustainable building materials.

Whereas demineralised water mainly induces dissolutions, natural water can cause, in addition, the precipitation of a calcium-rich crystalline layer, depending on the calcium content, e.g., the calcium hardness of water, whose consequence on leaching kinetics has yet to be made clear. Indeed, initial studies showed that the calcite layer clogs the porosity at the interface between the cementitious matrix and the external environment, thus protecting the cement paste [6,12,13]. However, other studies found that this barrier moves the diffusion front toward the core of the specimen [9,14]. Additionally, calcite formation would increase the gradient of calcium concentrations and, thus, enhance structural deterioration. Moreover, the exact boundaries of the clogging domain, mainly depending on pH, water carbonates, and calcium concentrations, need to be better defined because of the uncertainty concerning the nucleation, growth, and microstructure of calcite formation [10]. The stability of the calcite layer is also discussed, and the shape of the mass loss vs. time curve can illustrate this. If close to a parabolic function, the outer layer is stable, and conversely, a linear shape suggests it is continuously removed [14]. The formation of calcite is, thus, directly linked with the water hardness, and the latter affects the effective diffusion coefficient [15] and leaching mechanisms. To sum up, research works have different views on the long-term consequence of the calcite formation at the interface with leaching water; some have shown this would greatly reduce or stop cement decalcification, whereas others argue that decalcification would continue as long as a concentration gradient was maintained between the external solution and the pore solution via renewal of external solution [10], the difference being the ability of calcium carbonate to clog the porosity. Some research works suggest it would depend on the flow of leaching water [4].

The durability of concrete structures is taken into account in standards such as EN 206+A2 [16] by considering some parameters of the environment independently, such as pH or the carbonate ion concentration of water, but the standard does not take into account calcium ions, which are key ions in the leaching phenomena of hydration products such as portlandite and C-S-H. These indicators allow durability concerns to be considered during the concrete mixture design stage but do not fully consider the complex interaction between cement-based materials and water, as described before. Moreover, all the ions present in water have a significant influence on the equilibrium between the pore solution of concrete and aggressive water [15]. For example, aluminium and iron are stabilising elements for the calcite outer layer by entering the structures of C-S-H and modifying their properties [14]. Also, the characteristics of the outer layer change when the pH decreases, of which the calco-carbonic equilibrium of the water is a major governing parameter. Numerous experimental protocols have been defined to reproduce the leaching phenomenon of cement-based materials [17,18]. The aim is to maintain the initial conditions of temperature, pH, and chemical composition of the leaching water. Some procedures consist of the continual renewal of the leaching solution using a pump [19]. Nitrogen gas is used to avoid any risk of carbonation. In the Leachcrete test, the heating mantle allows the evaporation of the water, which is thus deionised. Then, the water is cooled before returning to the container where the samples are immersed [17]. To maintain the leaching over time, samples can also be immersed into a large container, and no renewals are made [8,19,20,21,22], but the risk is that the deterioration ends up slowing down and stopping after a few days or weeks. In order to accelerate deterioration, numerous studies have been performed using nitrate ammonium solutions [1,23]. Another possibility is renewing periodically the leaching solution and controlling the pH in the interval [8,24,25]. Studies often consider the composition of water in order to combine different attack mechanisms, e.g., an external sulphate attack and leaching [26,27,28], or focus on specific in situ cases [1,29]. Therefore, to the authors’ knowledge, there is currently no consensus on the type of performance-oriented leaching test that can easily be used with mineral water to obtain useful insights into water-induced cementitious material degradation. Moreover, the difference between mineral water types, e.g., soft and hard water, on leaching has yet to be reported using such tests.

The study presented in this paper addresses this issue as it combines the description of the aggressiveness of natural waters based on the Langelier saturation index (LSI), which is aimed to describe the water’s aggressiveness based on the calco-carbonic equilibrium, with a comprehensive investigation on the leaching mechanism of studied materials. This global approach includes the aggressive environment and the specimens, using both macroscopic indicators and microstructural characterisations through thermogravimetric analyses (TGAs), X-ray microtomography, and 3D microscope observations. Portland cement mortar specimens were exposed to two natural bodies of waters with LSIs of LSI+0.5 and LSI−1.0 corresponding to scaling (hard) with respect to aggressive (soft) water and demineralised water. The water was regularly renewed with two renewal criteria, and the pH was controlled by titration with 0.5 M nitric acid to allow the monitoring of degradation conditions. The evolution of the composition of the water was evaluated by calcium titration with EDTA. Sample deterioration was assessed by monitoring their mass in air and water and via TGA measurements along with 3D microscopic and X-ray microtomography observations. These indicators showed that two mineral waters lead to different degradation mechanisms depending on their nature determined based on the Langelier index.

## 2. Materials and Methods

### 2.1. Mortar Samples Preparation

The natural leaching of concrete being relatively slow, materials were selected in order to accelerate the sample deterioration during laboratory tests. This enabled the analysis of the phenomena and allowed differences to be observed between samples placed in different environments.

CEM I Portland cement was selected as the binder [30]. Its chemical composition and phase composition are given in Table 1. Considering its low C_3_A content and high C_3_S content, it leads to a significant proportion of portlandite (16% of cement paste after 28 days), which is known to be vulnerable to leaching. In total, 0/2 mm of siliceous sand from Palvadeau, France, was used to minimise the interactions between aggregates and the leaching solution. The sand density was equal to 2.61 kg/L.

Cylindrical mortar samples of a 0.6 water-to-cement ratio were prepared according to the NF EN 196-1 standard [31]. While cement pastes have the advantage of being more adapted for chemical analysis or observations, it is not representative of the concrete used for construction. It is important to take into account the microstructure and the specific properties of the interface between the paste and aggregates. Some research works have shown that aggregates are likely to induce cracking [32]. Nguyen has also observed that the deteriorated depth during leaching depends on the presence or lack of aggregates [33]. Kamali et al. compared the deteriorated depth due to leaching by demineralised water for CEM I, CEM III, and CEM V cement, and they showed that the water-to-cement ratio did not modify leaching main trends (ranking of the degradation speed) [18]. Thus, a high water-to-cement ratio has been chosen to foster leaching alongside CEMI.

Fresh mortar was poured in two layers into cylindrical moulds of 20 mm in diameter and 170 mm in height. Each layer was vibrated for 10 s on a vibration table. Samples were unmoulded after three days and then cured in lime-saturated water at 20 °C until the test started at the age of 28 days. Before leaching tests, both ends of the sample were covered by epoxy resin to obtain a two-dimensional deterioration. Leaching tests were started at the age of 60 days. At the same time, an initial porosity of 17.9% was measured using water saturation following a procedure derived from standard EN 13057 [34]. Three slices (2 cm) of samples were saturated under a vacuum. Samples were placed in a desiccator under a vacuum for 4 h, then underwater and a vacuum for 44 h. After being removed from the desiccator, the samples were weighed in water and then in air after being briefly dried on the surface. The samples were then dried in an oven at 105 °C. They were weighed every day until the difference in mass between two successive 24 h weigh-ins was lower than 0.05%.

### 2.2. Water Compositions

Two mineral waters and demineralised waters were used. Their chemical compositions are given in Table 2. Commercial mineral drinking waters were selected based on their calculated Langelier saturation indices of LSI+0.5 and LSI−1.0 corresponding to scaling with respect to aggressive water [35]. Demineralised water (UW) was used as a reference. Two samples were kept in lime-saturated water as reference samples (Ref).

The Langelier saturation index is defined as the difference between the measured *pH* and the *pH* for which the water reaches its calco-carbonic equilibrium, $pH_{s}$ (Equations (1) and (2)):(1)$$LSI=pH-pH_{s}$$
(2)$$pH_{s}=pK-pK_{s}+2\varepsilon +7.7-\log\left(TAC\right)-log(TCa)$$
where *TAC* is the total alkalimetric title, *TCa* is the calcium hardness, and pK the reaction constant of the bicarbonate ions deprotonation reaction at a given temperature:(3)$$HCO_{3}^{-}\to H^{+}+CO_{3}^{2-}$$
given the $pK_{s}$ of calcium carbonate formation as follows:(4)$$Ca^{2+}+CO_{3}^{2-}\to CaCO_{3}$$

Thus, the LSI gives an insight into the capacity of a given water type to enable the precipitation of calcium carbonate in the presence of bicarbonate and calcium ions. When LSI equals zero, no calcite can dissolve nor precipitate, whereas when it is positive (respectively negative), calcite can form (dissolve). LSI can give indications on the usability of water for agriculture [26,36] or its drinkability [37], and it is part of a set of indicators designed to characterise the aggressiveness of water in water supply networks.

### 2.3. Leaching Protocol

The test procedure illustrated in Figure 1 aims at reproducing the leaching of mortar by mineral waters, which were periodically renewed for six months. In each of the three water compositions, three mortar samples were immersed in 1.8 L of leaching water with magnetic stirring in an open system. The sample surface per leaching water volume ratio was 5.9 m^−1^. Control specimens were stored in lime-saturated water. During the experiment, a pH of 7.0 and a temperature of 20 °C were controlled and kept constant using nitric acid and a temperature regulator, respectively.

Two different criteria were used for the renewal of leaching solutions, as detailed in Table 3. The first one (1) was based on the evolution of LSI during leaching. The evolution of the chemical compositions of water during leaching led to an evolution of water aggressiveness. The renewal of the leaching solution was expected to ensure aggressive (respectively scaling) waters will keep being aggressive (with respect to scaling). The ranges of Langelier saturation index values are defined for each type of water. When the Langelier saturation index value went out of the range, the solution was renewed. The second criterion (2) was temporal; the solution was renewed weekly.

### 2.4. Macroscopic Measurments

During each renewal of leaching mineral water, mortar sample mass in air and water were measured. They were wiped until the saturated surface dried state to measure the mass in air. Volume was calculated from both measurements (Equation (5)).
(5)$$∆V=\frac{∆M_{air}-∆M_{water}}{\rho_{w}}$$
where ∆V is the gain of volume, $∆M_{air}$ is the gain of mass in the air, $∆M_{water}$ is the gain of mass in water and $\rho_{w}$ is water density.

Macroscopic chemical measurements were also performed: the calcium hardness (TCa) was obtained via colorimetric tests from ORCHIDIS (Champigny sur Marne, France) on the renewed leaching water using a dropper. It represents the quantity of calcium and magnesium ions in the solution. In the leaching test, there was no exchange of magnesium ions between the mortar and the leaching water. Thus, the variation in calcium hardness in water was used for calcium content monitoring in this study. The measurement unit is the French degree (°f). One French degree is equivalent to 4 mg of calcium or 2.4 mg of magnesium per litre of water.

### 2.5. Microscopic Test and Imaging

In order to identify solid phases, thermogravimetric analyses (TGA) were performed using a Netzsch STA 409 PC Luxx (Selb, Germany) device on crushed powders of the specimen after the leaching test. Mass variations in the sample were measured as it was heated. The identification of phases was taken from their transformation temperatures. In total, 120 mg of powder was heated under a nitrogen atmosphere with a constant heating rate of 20 K/min up to 900 °C. The analysis was performed from a complete slice of a mortar cylinder (two tests per slice for repeatability). A calibration was performed in order to eliminate measurement defects.

The surface of mortar samples was observed after degradation using a 3D microscope. Three-dimensional maps of three lines were obtained along the height of each specimen using a Hirox RH2000 3D microscope (Tokyo, Japan) by merging hundreds of images evenly spaced along the section [38]. A magnification of ×50 was chosen according to the standard suggestion, leading to a final horizontal resolution of 3.13 μm/m. Equally spaced images were acquired along the Z-axis to infer the height of every pixel in the image, as described in [39]. Light intensity was manually adjusted to mid-range values so as to limit the potential reflection and subsequent blurring due to calcite precipitates [40]. After image acquisition and processing, the arithmetical mean roughness Ra (integral over the profile of the absolute values of deviations between the mean line of the profile and the measurement curve) and the 10-point mean roughness Rzijs (the average value of the absolute values of the 5 highest heights and 5 lowest depths) were calculated.

High-resolution Zeiss Xradia MicroXCT-400 X-ray microtomography (Oberkochen, Germany) was also performed to observe the deterioration in-depth due to leaching tests, as illustrated in Figure 2, using a tension of 120 kV. Each image was exposed for 8 s from a −180° angle to a +180° angle. The reassembling of the 1080 images was performed thanks to the software XMReconstructor version 10.7.3245. The final resolution was 11.68 µm/voxel. In the end, an equalisation was performed to enable a simple comparison of the different images and an easier visualisation of the deteriorated front using GIMP software (https://www.gimp.org/, 15 January 2024), as illustrated in Figure 3.

## 3. Results and Discussion

### 3.1. Visual Observation of the Degraded Mortar Samples

#### 3.1.1. 3D Microscope Results

Visual observations confirm the loss of solid phases at the surface of mortar samples submitted to the leaching test with the same water renewal frequency (Table 4). A remarkable visual difference can be observed between the aspects of samples exposed to demineralised, mineral, and lime-saturated water, as illustrated in Figure 4. From the photograph, it can be concluded that the surface roughness is considerably increased due to leaching since the reference sample surface is very smooth with only a limited number of air bubbles, while the surface of leached samples is rough, which is in agreement with onsite observations. Sand particles are clearly visible on the demineralised and mineral water-exposed samples.

Roughness measurements along the sample height using the 3D microscope confirm these visual observations. The average roughness values were 12.9, 10.5, 8.9, and 2.9 for samples UW, LSI+0.5-2, LSI−1.0-2, and the reference specimen (kept in lime-saturated water), respectively. (Table 4). Three-dimensional measurements accurately quantify the degradation impact on the sample’s surface, and a clear difference is visible between specimens subjected to leaching and lime-saturated-exposed specimens. The surface degradation amplitude follows the order UW > LSI+0.5-1 > LSI−1.0-1 >> Ref. The same order was confirmed by Rzijs, which means that deeper surface degradations are caused by demineralised water, followed by hard water. Demineralised water thus induces the most impactful surface degradation and degradation that is significantly more pronounced than mineral water-exposed samples. The roughness of the hard water-exposed sample is slightly higher than that of the soft water-exposed specimen, primarily due to zones with significant deterioration. In contrast, other zones are covered with a thin white deposit.

#### 3.1.2. X-ray Microtomography Results

After 300 days of the leaching test, the mortar samples exposed to mineral water were further characterised through X-ray microtomography. Surface deterioration was observed on mortar submitted to the test via an increase in porosity, which was not observable in the reference mortar sample immersed in lime-saturated water. The leached depth was estimated based on the location of the degradation front between the sound zone and the zone with higher porosity, as illustrated in Figure 5. The average of 24 measures gives a deterioration depth equal to 2.7 mm for LSI−1.0-2 (standard deviation of 0.25 mm) and 1.6 mm for LSI+0.5-2 (standard deviation of 0.31 mm). The soft water caused a deeper deterioration of the mortar sample than the hard water. The calcium concentration gradients between the cementitious matrix and the soft water were actually higher than hard water. A denser layer at the edge of the sample immersed in LSI+0.5-2 was observable thanks to image processing. This confirmed the 3D microscope observations, and it was likely a calcite layer. Based on both complementary imaging techniques, it can then be concluded that demineralised water induces the most substantial degradation, which has already been previously reported. Mineral water also induces severe degradation; hard water degradation is mainly localized close to the surface with particularly depredated spots besides thin calcite-covered areas, and soft water degradation is visually similar to demineralised water, yet slower.

### 3.2. Evolution of Monitoring Parameters

The actual testing conditions are compared and discussed in this section. The renewal frequency is an aggressiveness factor [18]. The renewal of solutions is necessary in order to keep waters in a chosen range of aggressiveness. The frequency depends on the renewal criteria. In Figure 6, the total number of renewals is plotted against the time for the leaching test performed according to the first renewal criterion. Except during one week for UW and LSI−1.0-1, the renewal frequency was always equal to or higher than once a week. The frequency was remarkably high at the beginning of the leaching tests (9, 7, and 8 renewals during the first week for LSI+0.5-1, LSI−1.0-1, and UW resp.). After one month, the average frequency was lower: 2.7 for LSI+0.5-1, 1.3 for LSI−1.0-1, and 1.1 for UW. UW had almost one renewal per week, as for tests performed according to the second criterion, and thus, it was comparable after one month. LSI−1.0 water had a slightly higher renewal frequency than UW, while LSI+0.5 water needed to be renewed with a significantly higher frequency, both controlled at a pH of 7. This can be attributed to the buffer effect [15].

As part of performance-based specifications, the deterioration indicators must be accessible, i.e., simple and easy to implement while giving direct information on the deterioration process. Several accessible indicators are presented in this section.

The following mass variations have been deduced from the difference between control and leached samples, and thus, they represent only the impact of leaching without the effect of hydration. Mass was monitored weekly during the leaching test, as illustrated in Figure 7a. During the first period, the mass variations in samples showed the same linear evolution, with a loss of 1.6% after 130 days for specimens LSI−1.0-2, LSI+0.5-2, and UW. During the second period, while the mass of samples in UW was still linearly decreasing, the mass loss slowed down for the samples exposed to mineral waters. LSI+0.5-2 lost 0.22% within the 130 following days, and LSI−1.0 lost 0.6%. This mass loss slowdown could be attributed to calcite formation at sample surfaces clogging porosity, thus reducing ion diffusivity through this layer. The time of evolution to weight loss follows a parabolic function, which suggests the presence of a stable protective layer, whereas when the latter is removed continuously, the function adopts a linear shape [14,41].

The mass variation in water due to leaching over time is given in Figure 7b. The data confirm that whereas the demineralised water induces a linear loss of mass in water all over the immersing time, a slowdown of the mass in water loss appeared for both mineral waters after around 130 days, but with a stronger effect for the hard water LSI+0.5-2. It is interesting to observe that the curves of mass in the water of the specimens exposed to mineral waters crossed during the leaching test. Before 85 days, hard water induced the same mass loss as the UW and the soft water. It might not seem logical at first, as the ionic gradient was smaller. One assumption to explain this trend is the quantity of nitric acid added to the beaker to control the pH. Due to the buffer effect, the hard water needed more nitric acid (30.4 mL per week for LSI+0.5-2 and 19.9 mL for LSI−1.0-2). As demonstrated by Pavlik [6], the concentration of acid had a stronger influence than the pH. The higher concentration of carbonate ions could also explain the aggressive character of LSI+0.5-2 at the beginning. Calcite formation requires calcium ions and, thus, could act as a calcium pump and accelerate leaching [4]. This has also been highlighted by some numerical studies [42]. Then, the hard water would have created enough calcite to generate the clogging of porosity in the external layer, resulting in a stronger reduction in the deterioration rate than the soft water. Conversely, if the calcium concentration is lower, the creation of calcite cannot lead to the clogging of porosity [4].

Figure 8 shows the mass variations in samples in soft and hard water related to both renewal criteria. They can be compared with the graph in Figure 7. The mass variation due to soft water may initially appear counterintuitive. First, LSI−1.0-1 was renewed with a higher frequency than LSI−1.0-2, which resulted in a higher concentration gradient between the cementitious matrix and environmental water. Moreover, the objective of the renewal criterion of LSI−1.0-1 was to keep the water aggressive, according to the LSI. This is why higher deteriorations could be expected in the first approach, whereas this is not the case. LSI−1.0-1 actually caused a slightly lower mass loss than LSI−1.0-2. It can be assumed that water cyclically changing from an aggressive to scaling LSI induces a more severe degradation of concrete than water kept in an aggressive state [43]. This, alongside visual observations, could corroborate the idea that calcite formation acts as a calcium ion pump and, thus, accelerates the dissolution of calcium minerals as long as the clogging effect does not occur.

Figure 8b illustrates the global mass variations in LSI+0.5 according to both renewal criteria. On the opposite hand, LSI+0.5-1 produced more deterioration than LSI+0.5-2. LSI+0.5-1 was renewed about three times more than LSI+0.5-2 at a higher renewal frequency difference than between both soft waters. Renewals prevented the formation of a protective calcite layer at the surface of the sample. In addition, the mass gain of LSI+0.5-1 did not reach any plateau, even if it exhibited a parabolic trend. This layer was formed more slowly.

The chemical composition of water was monitored during leaching tests. The calcium hardness of water was measured and cumulated over time (Figure 9). Calcium ions came from the dissolution of cement hydration products and, thus, were a direct indicator of the leaching rate. The soft water and UW caused the same calcium loss, whereas the evolution of cumulated calcium hardness in hard water could be separated into two periods. First, the dissolution was faster than for other waters with a gain of 150 °f within 30 days. The slope of LSI+0.5 was even higher than UW, with a gain of 105 °f in 30 days. The renewal frequency of LSI+0.5 was actually higher than UW in the first 30 days. Then, it stopped increasing. No more calcium minerals seemed to be dissolved. These two different types of evolutions confirm the formation of calcite during exposure to hard water. Calcite is likely to clog the porosity and reduce the possible exchanges of ions between the cementitious matrix and the environment. Another explanation could be that the diffusion coefficient decreases when the water hardness increases [15]. It can be noticed that the mass loss with UW was not linear with the square root of time. It can be influenced by the acid concentration and Fick’s law [44]. Thus, when the kinetics are not linear with the square root of time, it means either the diffusive coefficient has changed or the phenomena cannot only be described by diffusion. Therefore, it can be influenced by the evolution of porosity. LSI+0.5-2 induced the creation of a calcite layer at the surface of the mortar sample, which could lead to a decrease in the diffusion coefficient.

### 3.3. Analysis and Correlation between Macroscopic Indicators

The following volume variations were deduced from the difference between control and leached samples. Volume variations obtained during leaching tests performed according to the second renewal criterion are presented in Figure 10 with those obtained with UW.

It can be observed that during the first 30 days, volume loss differed. The hard water caused the same volume loss as UW (about 0.7% of loss). Still, it is noteworthy that LSI+0.5-2 was renewed much more often than UW. During the first week of the leaching test, UW was renewed eight times; thus, the samples were exposed to more aggressive conditions, as previously discussed. However, the specimens in soft water showed a lower loss of volume (0.4% in 30 days). During the second period, their volume losses followed the same linear evolution until 100 days. Then, UW led to the same evolution, whereas mineral waters induced a gradual slowdown of the volume loss. The gap between them was small. The fact that mass loss curves crossed rather than volume curves was noticed. This suggests that the dissolution and precipitation phenomena involved different phases and that the deterioration mechanisms were not the same.

In Figure 11, volume variations are plotted against mass variations for water renewed with a frequency close to once a week. The samples in hard water and UW evolved in a similar way, just under the first bisector. For the samples in soft water, the loss of volume was higher than the loss of mass. This graph underlines how soft water produces a greater loss of volume than mass, which could be explained by a low density of dissolved phases and a higher density of precipitated phases, with portlandite density being 2.2 g/L, while the calcite one is 2.7 g/L.

The calcite and portlandite contents of cross-sectional slices of mortar samples were assessed by TGA, as illustrated in Figure 12. Calcite actually formed during the leaching with both tested mineral waters (0.7% for the hard water and 0.5% for the soft water). The portlandite content was higher in the mortar immersed in hard water (2.0%) than in soft water (1.6%), testifying to its higher dissolution in soft water. It also indicates the reduction in dissolution when the calcite layer develops.

In Figure 13, cumulated calcium hardness variations were plotted against mass variations during the leaching tests performed on mineral waters. Three different evolutions were observed. The demineralised water shows the simplest evolution as follows: the leaching of calcium from the cementitious matrix led to mass loss. Mineral waters induced different evolutions. LSI+0.5-2 first showed more calcium release than UW for a given mass loss, and then it slowed down until the calcium release stopped, whereas the evolution of mass did not. It might be the effect of calcite formation acting as a calcium pump. The soft water first shows a linear evolution close to UW, and then the calcium release evolves faster than the mass loss. This might be explained by the fact that dissolution was more pronounced in mortar attacked by soft water. When leaching occurs, portlandite is first dissolved before monosulfoaluminates [5]. This evolution could allow the following new hypothesis to be formulated: due to the lower density of monosulfoaluminates than portlandite, it might induce a higher loss of mass, but the molarity is higher (1 mole of dissolved monosulfoaluminates frees 4 moles of calcium ions); thus, it has a higher effect on calcium hardness and more monosulfoaluminates might have been dissolved for LSI−1.0-2. This conjecture is yet to be confirmed.

The experimental measures presented so far are accessible in a comparative performance-based approach. They give distinctive profiles depending on the type of water. It is remarkable that the different profiles are very substantially marked, considering that the waters from this study, despite possessing two different LSIs, are both mineral waters. The water purification process involves different steps where water reaches LSIs that are significantly lower than −1.0. The indicators presented in this study are, thus, completely appropriate to characterise these types of degradation. Nevertheless, each indicator is influenced by several elementary phenomena, which can compensate for themselves, such as the dissolution or precipitation of calcium minerals, for instance.

## 4. Conclusions and Perspectives

The leaching of Portland cement-based mortar was induced by waters with different calcium concentrations and buffer capacities: demineralised, soft, and hard water. The Langelier saturation index (LSI) was aimed to describe water aggressiveness based on the calco-carbonic equilibrium. Portland cement mortars were subjected to leaching for 10 months in controlled conditions with constant pH. After various degradation durations, the samples were removed from the water and observed using a 3D microscope and X-ray microtomography. Macroscopic indicators, such as mass, volume, and cumulated variations in calcium ion concentrations, were monitored, and TGA analyses helped determine the nature of the precipitates at the surface of the samples. The main findings can be summarised as follows:Mineral water-induced leaching strongly differs from demineralised water-induced leaching, as confirmed by visual analysis, a 3D microscope, and X-ray microtomography analysis. Even though demineralised water-exposed mortars are more degraded than mineral water-exposed samples, the surface roughness of mineralised water-leached mortar samples was at least three times higher compared to lime-water-exposed reference samples. The roughness of leached specimens was 18% higher in hard water than in soft water. The degradation depth of soft and hard water-exposed specimens was 2.7 mm and 1.6 mm, respectively, after 300 days of the accelerated leaching test.The macroscopic indicators showed the strong influence that the nature of the water has on the type of degradation. The mass loss at 200 days was −2.16% for samples in demineralised water, −2.01% for samples in soft water, and −1.73% for samples in hard water.It was also observed that different mineral waters did not induce the same chemical mechanisms. Dissolution and precipitation reached a plateau after a few weeks in hard water, whereas soft water induced a constant rate of dissolution and precipitation. The cumulated calcium hardness variation in hard water was 69% lower than in soft water.Calcite formation at the surface of cementitious materials acts as a calcium ion pump, triggering ion propagation from the cementitious matrix towards the external layer of samples exposed to water. This phenomenon slows down when enough calcite is created to clog the open porosity of the mortar sample, reducing exchanges between the environment and the pore solution.

This study allows for a better understanding of the leaching mechanism. The results presented in this study show that considering the LSI of mineral water is a simple yet key factor that can help differentiate soft and hard water aggressiveness. The results presented herein might help understand the long-term behaviour of strategic mineral water-exposed structures such as water treatment plants and dams. It is worth mentioning that the mineral waters used in this study have relatively close LSIs but induce different leaching mechanisms. Therefore, it would be interesting to investigate whether the mechanisms defined in this study correspond to a given LSI or whether similar behaviours can be observed within the entire soft/hard water category.

## Figures and Tables

**Figure 1 materials-17-01548-f001:**
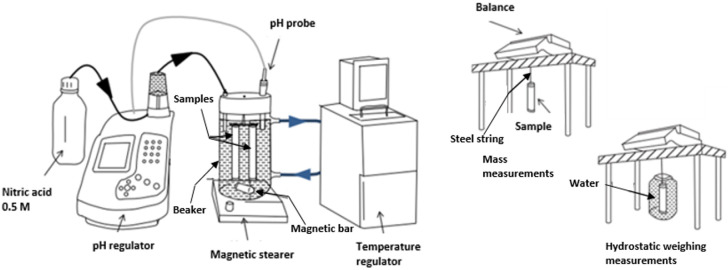
Experimental set-up for mortar leaching (**left**) and dimensional measurements (**right**), adapted from [30].

**Figure 2 materials-17-01548-f002:**
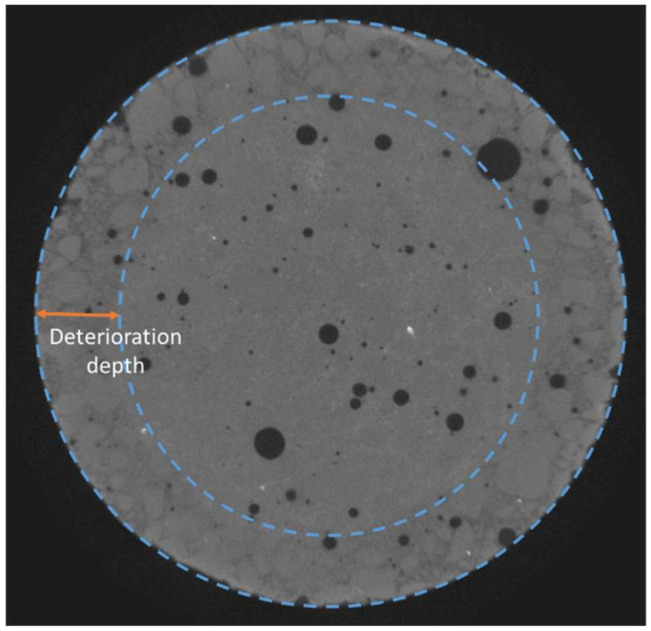
Deterioration depth measurement methodology (diameter of sample: 2 cm).

**Figure 3 materials-17-01548-f003:**
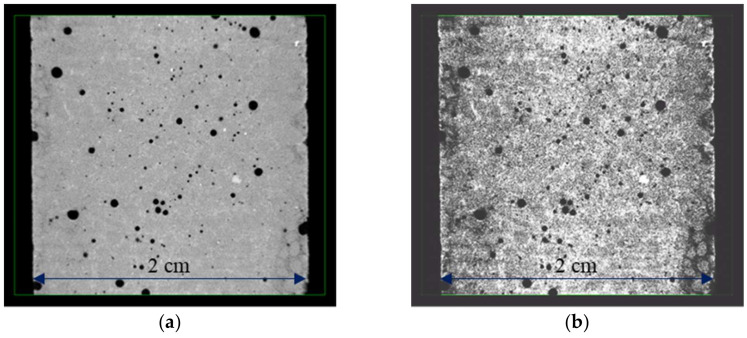
Visualisation of the degraded area: (**a**) sample picture obtained using microtomography, and (**b**) the same picture equalised using GIMP software (https://www.gimp.org/, 15 January 2024).

**Figure 4 materials-17-01548-f004:**
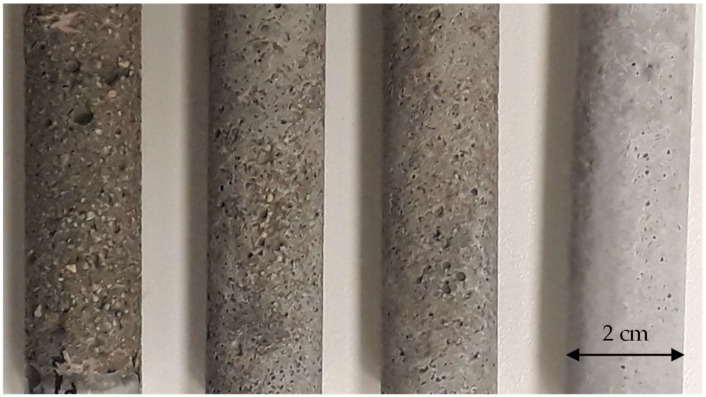
Samples after the immersion period (from left to right: immersed in demineralised water, hard water LSI+0.5-2, soft water LSI−1.0-2, and lime-saturated water).

**Figure 5 materials-17-01548-f005:**
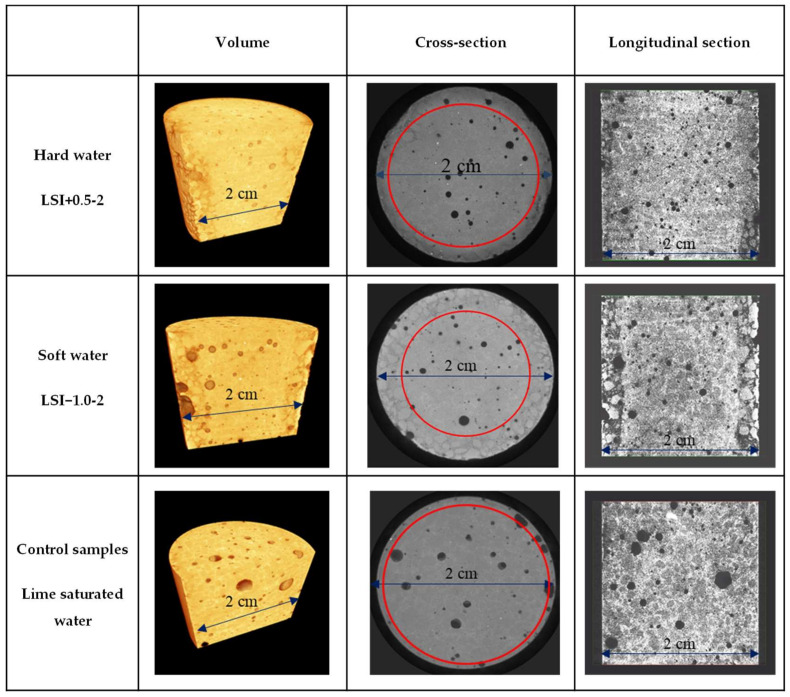
Images of mortar samples obtained using microtomography after 300 days of leaching test with a weekly solution renewal. The red circles distinguish the leached depth from the sound zone.

**Figure 6 materials-17-01548-f006:**
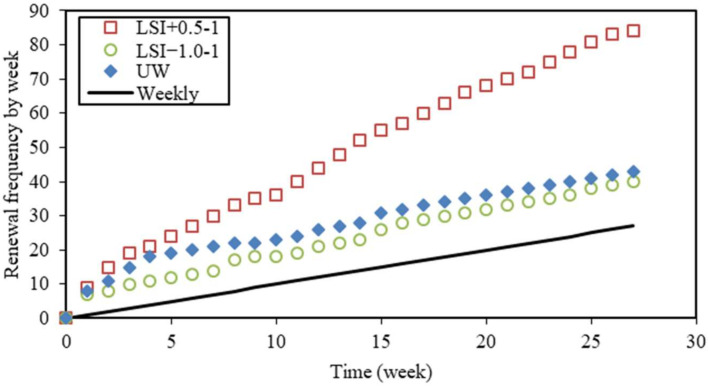
Cumulated number of solution renewals (LSI+0.5 means water with Langelier saturation index of 0.5, LSI−1.0 means water with Langelier saturation index of −1.0, and UW means demineralised water).

**Figure 7 materials-17-01548-f007:**
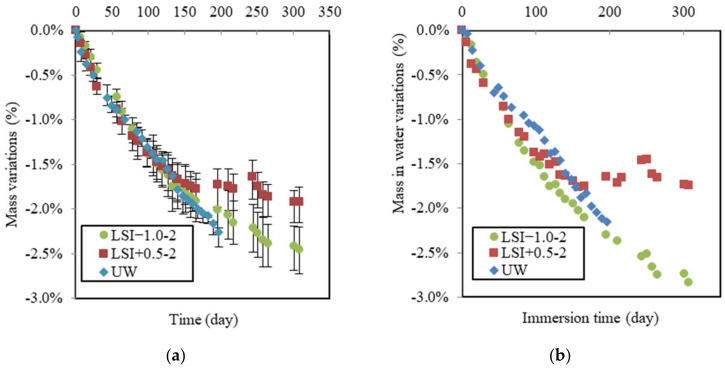
Mass gain (**a**), and mass in water (**b**) due to leaching following the second renewal guidelines (renewal every week).

**Figure 8 materials-17-01548-f008:**
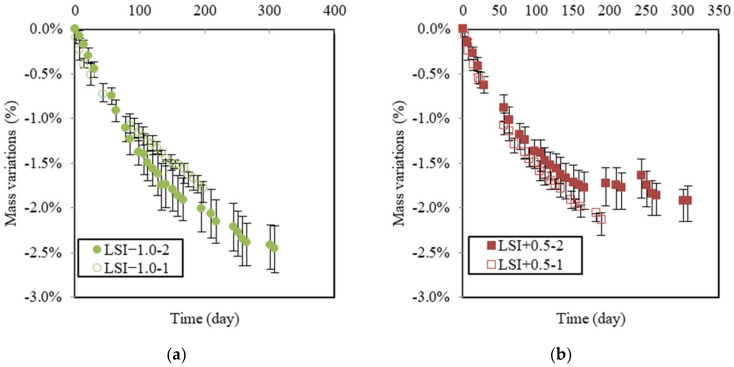
Mass gain due to leaching for soft water (LSI−1.0-1 means Langelier saturation index of −1.0 and renewal following acid titration; LSI−1.0-2 means Langelier saturation index of −1.0 and weekly renewal) (**a**). For hard water, (LSI+0.5-1 means Langelier saturation index of +0.5 and renewal following acid titration; LSI+0.5-2 means Langelier saturation index of +0.5 and weekly renewal) (**b**).

**Figure 9 materials-17-01548-f009:**
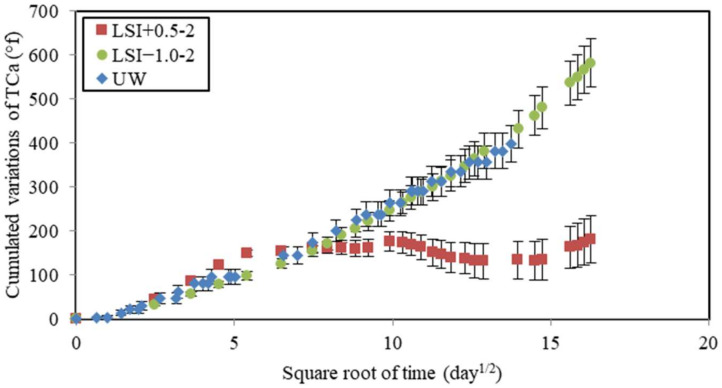
Calcium hardness variations cumulated over time as a function of the square root of time during leaching tests performed on mineral waters with a weekly renewal and demineralised water.

**Figure 10 materials-17-01548-f010:**
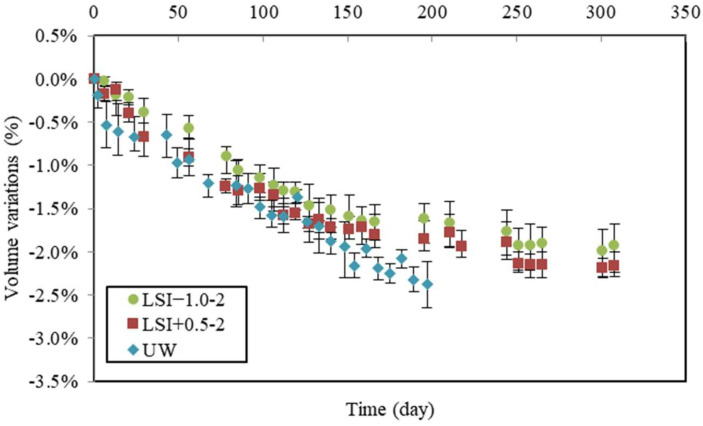
Volume variations due to leaching for the second renewal guidelines (weekly renewal) compared to mass variations in demineralised water.

**Figure 11 materials-17-01548-f011:**
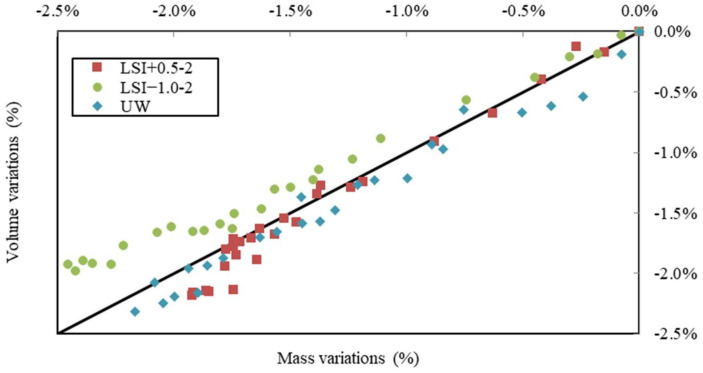
Volume variations vs. mass variations due to leaching performed on mineral waters with a weekly renewal and on demineralised water.

**Figure 12 materials-17-01548-f012:**
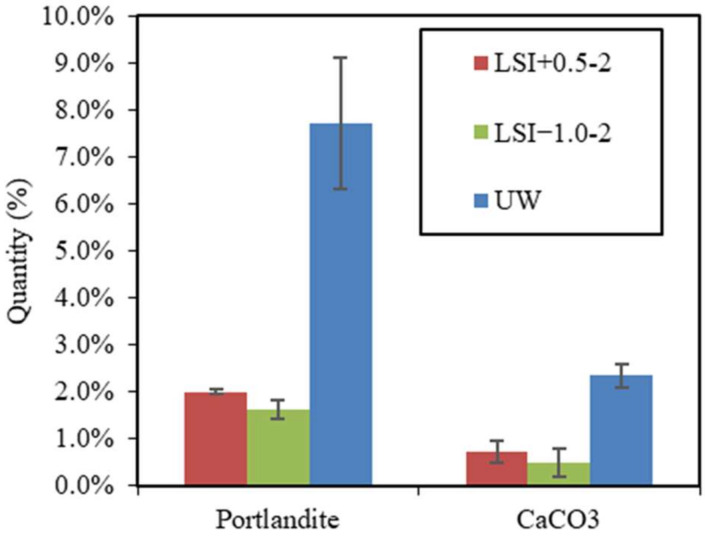
Portlandite and calcite quantities in mortar samples after 300 days of leaching by LSI+0.5-2, LSI−1.0-2, and UW.

**Figure 13 materials-17-01548-f013:**
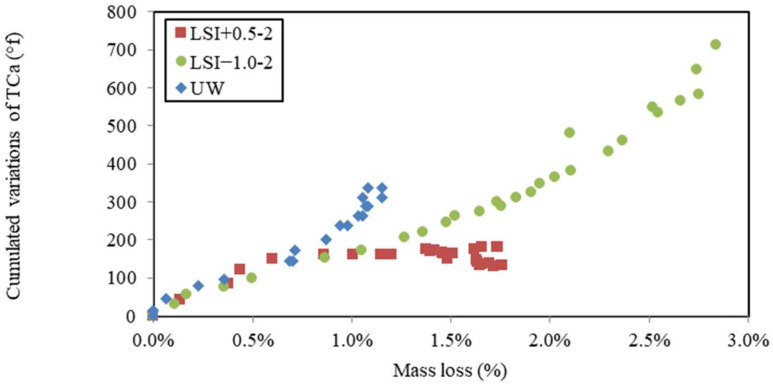
Calcium hardness variations cumulated over time according to gain of mass during leaching tests performed on mineral water according to weekly renewal and demineralised water.

**Table 1 materials-17-01548-t001:** Chemical and phase compositions of CEM I cement.

Chemical Composition (wt. %)				Phase Composition (wt. %)	
SiO_2_	20.11	Na_2_O	0.21	C_3_S	76.6
Al_2_O_3_	2.63	P_2_O_5_	0.09	C_2_S	10.9
Fe_2_O_3_	4.40	SrO	0.03	C_3_A	0
TiO	0.15	S^2−^	<0.02	C_4_AF	12.5
MnO	0.07	SO_3_	2.34	Anhydrite	3.7
CaO	64.15	Na_2_O total eq.	0.52	Calcite	8.1
MgO	1.02	Ignition loss	3.95	Portlandite	2.1
K_2_O	0.47			MgO	0.9

**Table 2 materials-17-01548-t002:** Chemical composition of tested waters (LSI+0.5 means water with a Langelier saturation index of 0.5, LSI−1.0 means water with a Langelier saturation index of −1.0 and UW means demineralised water).

	LSI+0.5	LSI−1.0	UW
Calcium (mg/L)	220	13	<0.4
Carbonates (mg/L)	<3	<3	<3
Potassium (mg/L)	2.2	6.3	<0.2
Magnesium (mg/L)	39	8.1	<0.2
Sodium (mg/L)	4.6	4.6	<0.4
Sulphates (mg/L)	385	8.95	<5
Chlorides (mg/L)	7.97	16.3	<1.0
Aluminium (mg/L)	<2	<2	3.0
Nitrates (mg/L)	3.8	7.9	<0.5
pH	7.8	7.5	6.2
LSI	0.5	−1.0	/

**Table 3 materials-17-01548-t003:** Renewal guidelines for leaching tests.

Name	Water	Renewal Criteria
LSI+0.5-1	Hard water	Langelier saturation index between −0.7 and 0.5
LSI−1.01	Soft water	Langelier saturation index between −1.7 and −0.7
UW	Demineralised water	Langelier saturation index lower than −1.7
LSI+0.5-2	Hard water	Weekly
LSI−1.0-2	Soft water	Weekly

**Table 4 materials-17-01548-t004:** Surface parameters of the leached mortars (Ref: control specimens).

Name	Ra (µm)	Rzijs (µm)
UW-1	12.9 ± 1.5	34.2 ± 5.5
LSI+0.5-2	10.5 ± 1.5	28.3 ± 5.5
LSI−1.0-2	8.9 ± 0.9	24.0 ± 2.8
Ref	2.9 ± 0.8	8.0 ± 2.3

## Data Availability

Data are available on request.

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
