# Peer review of "Experimental Investigation on the Effects of Mineral Water Composition on the Leaching of Cement-Based Materials"

_materials, 2024, doi:10.3390/ma17071548_

Round 1
Reviewer 1 Report
Comments and Suggestions for Authors
The presented manuscript focuses on the effect of water mineralization on the degradation or leachability of cement mortars and cement composites.
The submitted manuscript shows signs of a standard scientific work, the introduction to the problem is carefully handled, the materials and methods are appropriately described and the results are discussed. The conclusions follow the discussion and summarize the results obtained.
I have only a minor comment on the manuscript; it would be useful to add the manufacturer and country of the instruments used in sections 2.4 and 2.5
I recommend to continue in this direction and to further investigate the effect of water mineralization on the degradation of structures in contact with it.
The authors of the paper could improve the abstract , in which they could include specific results at least in percentage terms.
Furthermore, I suggest to complete the conclusion in which the authors would summarize the results obtained with specific values
Reviewer 2 Report
Comments and Suggestions for Authors
The purpose of the article is interesting. The research contribution is presented. The manuscript presents a clear description of the methodology and an adequate results discussion. I have some suggestions.
1. 1. The topic Introduction contextualizes the topic in order to justify the study as well as describe the contribution of the research to advancing the frontier of knowledge. After contextualization, the objectives must be presented, as well as a general description of the materials and methods used. Therefore, I suggest rewriting the last paragraph of the Introduction.
3. Tables and figures must be within the margins.
4. Figure 1: Words are not clear.
5. REF is probably a Reference specimen but is not described in the text.
6. In figures and tables, abbreviations must be described in the captions: figures and tables must be self-explanatory.
7. The conclusions could be more summarized, focusing on the main results.
Reviewer 3 Report
Comments and Suggestions for Authors
In this paper, leaching of Portland cement-based mortar was induced by two types of drinking water with different calcium content and buffer capacity in controlled conditions.
It is an interesting study concluding that the two mineral waters studied lead to different degradation mechanisms depending on their nature. The paper is well written, to all parts of it, from addressing the issue in introduction to the conclusions and perspectives. I suggest the paper to be published.
I provide provide some additional specific comments on the manuscript to be taken into account:
- Why were the specimen cured into lime-saturated water? The age could be written at line 139.
- What are mixing proportions for sand?
- Give more information about the method used for initial porosity (standard etc)
- line 263: “The leached depth was estimated based on the location of the degradation front between the sound zone and the zone with higher porosity, as illustrated in Fig. 5”. Figure 5 does not clearly show this.
-line 263: “…the reference mortar sample immersed in lime-saturated water”. Do you mean the Demineralized water here?
